# Dietary Barley Leaf Mitigates Tumorigenesis in Experimental Colitis-Associated Colorectal Cancer

**DOI:** 10.3390/nu13103487

**Published:** 2021-09-30

**Authors:** Daotong Li, Yu Feng, Meiling Tian, Xiaosong Hu, Ruimao Zheng, Fang Chen

**Affiliations:** 1Health Science Center, Department of Anatomy, Histology and Embryology, Peking University, Beijing 100191, China; lidaotong@bjmu.edu.cn; 2National Engineering Research Center for Fruit and Vegetable Processing, Key Laboratory of Fruits and Vegetables Processing, College of Food Science and Nutritional Engineering, Ministry of Agriculture, Engineering Research Centre for Fruits and Vegetables Processing, Ministry of Education, China Agricultural University, Beijing 100083, China; fengyu9459@163.com (Y.F.); tml0214@163.com (M.T.); huxiaos@263.net (X.H.)

**Keywords:** barley leaf, colitis, colorectal cancer, intestinal barrier function, gut microbiota

## Abstract

Dietary barley (*Hordeum vulgare* L.) leaf (BL) is a popular functional food known to have potential health benefits; however, the effect of BL in colorectal cancer prevention has not been examined. Here, we examined the role of BL on the prevention of colorectal carcinogenesis and defined the mechanism involved. BL supplementation could protect against weight loss, mitigate tumor formation, and diminish histologic damage in mice treated with azoxymethane (AOM) and dextran sulfate sodium (DSS). Moreover, BL suppressed colonic expression of inflammatory enzymes, while improving the mucosal barrier dysfunctions. The elevated levels of cell proliferation markers and the increased expression of genes involved in β-catenin signaling were also reduced by BL. In addition, analyses of microbiota revealed that BL prevented AOM/DSS-induced gut microbiota dysbiosis by promoting the enrichment of *Bifidobacterium*. Overall, these data suggest that BL is a promising dietary agent for preventing colitis-associated colorectal cancer.

## 1. Introduction

Colorectal cancer has become one of the most common malignancies in the world [1]. As one common type of inflammatory bowel disease (IBD), ulcerative colitis features chronic inflammation in the large intestine, which is an important risk factor closely related to the occurrence of colorectal cancer [2,3,4]. Moreover, it is estimated that more than 1 million patients are diagnosed with colorectal cancer each year, and more than 20% of patients die of the disease [5,6]. Recently, several anti-inflammatory drugs have been demonstrated to effectively ameliorate the incidence of colorectal cancer [7]. However, the efficacy of such agents is not satisfactory and may also trigger severe life-threatening side effects [8]. Hence, it is still necessary to find efficacious and safe treatment strategies and measures to lower the risk of colorectal cancer.

The underlying pathogenesis of the disease remains unclear, and the initiation and development of colorectal cancer may be associated with multiple etiological factors [9]. Chronic inflammation in the initial stages of cancer is caused by the infiltration of various types of inflammatory cells [10]. The inflammatory response is also crucial for the progression of carcinogenesis through mediating multiple signaling pathways involved in tumor cell proliferation, angiogenesis, and metastasis [6,11,12]. The aberrant accumulation of inflammatory cells and release of cytokines also damage the colonic mucosa and lead to the repeated injury of the intestinal barrier [5]. Hence, blocking the sustained activation of inflammatory signaling is an effective strategy in suppressing colon tumorigenesis [13].

As an important environmental factor, the gut microbiota is considered to play a role in regulating the tumorigenesis of colorectal cancer [14,15]. Dramatic disruption in intestinal microbial composition, termed microbiota dysbiosis, is frequently detected in individuals with colorectal cancer [16]. Since diet is an important factor affecting the gut microbiome, the emerging links between western dietary habits and colorectal cancer risks may also be attributed to the effects of a western diet on the intestinal microbiota [17]. For instance, the formation and development of colorectal cancer is considered to be triggered by dietary heme intake, which leads to a gut-bacteria-dependent intestinal hyperplasia [18]. In contrast, an increased intake of fruits and vegetables is beneficial to control colorectal cancer incidence through regulating the gut microbiota [19,20]. Thus, the development of effective dietary strategies targeting the gut microbiota can have great significance for the prevention of colorectal cancer.

Barley leaf (BL) is rich in natural nutrients and is also a functional food with various health-promoting activities [21]. We and others have shown that BL has protective effects against colitis through the regulation of gut microbiota [22,23,24]. However, evidence of dietary BL in the prevention of colitis-associated colorectal cancer is still lacking.

In this research, we evaluated the cancer-protective role of BL in an azoxymethane (AOM)/dextran sodium sulfate (DSS)-induced murine model of colorectal carcinogenesis. Our results demonstrated that the preventive efficacy of dietary BL on tumorigenesis occurred through attenuating inflammatory responses, enhancing mucosal barrier function, and suppressing microbiota dysbiosis.

## 2. Materials and Methods

### 2.1. Nutritional Composition of BL

The BL was prepared as previously described [23] and its nutritional composition is shown in Appendix A. The contents of nutritional components including protein, fat, soluble dietary fiber, insoluble dietary fiber, moisture, and ash were detected based on the National Standard Test Method by PONY (Beijing, China).

### 2.2. Animal Experimental Design

The C57Bl/6J mice (male, 5 weeks old) were raised under standard conditions with 12 h light/dark cycles. All protocols for animal experimental manipulations were carried out under the institutional guidelines and were approved by the Animal Ethics Committee of the Peking University Health Science Center.

The experimental protocols for the induction of the colorectal cancer mouse model was performed as previously described [24]. After adapting to the laboratory conditions for 7 days (Week 1), forty mice were randomly divided into four groups (*n* = 10). Mice in the control and AOM/DSS groups were fed a standard chow diet. Mice in the AOM/DSS + 2.5% BL and AOM/DSS + 5% BL groups were fed an isocaloric diet, supplemented with BL with a ratio of 2.5% or 5%. The detailed nutritional composition of the diet is provided in Appendix A. At Week 0, mice with colorectal cancer induction were injected intraperitoneally with 10 mg/kg of AOM. After one week, mice were then administered with 2.5% DSS in drinking water for one week. This was followed by 2 weeks of regular drinking water, and the treatment cycle was repeated for two more times. At Week 10, mice were sacrificed by cervical dislocation.

### 2.3. Evaluation of Disease Activity Index (DAI)

A clinical assessment of DAI was performed using an established colitis severity evaluation system, assessed from a combined score including weight loss, diarrhea, and rectal bleeding, and evaluated by two researchers in a blinded manner [25].

### 2.4. Evaluation of Colonic Tumors

After opening the colon longitudinally and washing the feces with saline, we examined the number and size of colon tumors of each mouse. According to their diameters, the tumors were divided into three groups: small tumors with a diameter of less than 1 mm, medium tumors with a diameter of more than 1 mm and less than 2 mm, and large tumors with a diameter of more than 2 mm. Tumor load was calculated as previously described [24].

### 2.5. Evaluation of Histological Severity

The colon tissue samples were fixed with 4% paraformaldehyde and were cut in sections (4 μm). Colonic slides were stained with hematoxylin and eosin (HE) to assess the extent of histological and crypt injury. In addition, colonic sections were also stained with Alcian blue (AB) to evaluate mucosal barrier damage by counting the goblet cells [23].

The histopathological scores of colonic lesions were measured by calculating the total scores of crypt damage, inflammation severity, and depth of injury from a previously established scoring system [26].

### 2.6. Measurement of Cytokine Production and Myeloperoxidase (MPO)

Collected colonic tissue samples were homogenized in radio-immunoprecipitation assay (RIPA) buffer (Solarbio, Beijing, China) and centrifuged. The protein concentration of the supernatants was measured. The cytokine levels in the colon, including interleukin (IL)-6, IL-1β, tumor necrosis factor (TNF)-α, and interferon (IFN)-γ, were quantified by commercial ELISA kits as described previously [23].

The level of inflammatory enzyme MPO was determined using a commercial ELISA kit (Nanjing Jiancheng, Nanjing, China).

### 2.7. Intestinal Epithelial Permeability Analysis

The barrier permeability of the intestinal tract was measured using the fluorescein isothiocyanate (FITC)-dextran (Sigma-Aldrich, Saint Louis MO, USA) method [23]. Before the assay, mice were fasted for 4 h, followed by the administration of FITC-dextran by gavage. After another 3 h, the blood samples of mice were collected and the fluorescence intensity of FITC-dextran in the serum was measured.

### 2.8. Immunohistochemistry and Immunofluorescence

Immunohistochemistry of colonic sections was performed as described previously [23]. After microwave antigen retrieval, the colonic slides were incubated overnight with primary antibodies of Ki-67 (Abcam, ab21700, Cambridge, UK) and proliferating cell nuclear antigen (PCNA) (Abcam, ab29, Cambridge, UK) at 4 °C. For immunofluorescence staining, colonic slides were incubated overnight with primary antibodies of β-catenin (Abcam, ab32572, Cambridge, UK) and Muc-2 (Abcam, ab272692, Cambridge, UK) at 4 °C.

### 2.9. Real-Time qPCR

The real-time quantitative PCR assay was used to quantify the changed expression of genes in the mouse colon. The total RNA from colonic tissues was extracted using Trizol (Thermo, Waltham, MA, USA). Detection of gene expression was determined by reverse transcription, cDNA synthesis, and qPCR, as previously described [23]. The expression of genes was calculated and normalized to Glyceraldehyde-3-phosphate dehydrogenase (GADPH). The information of primer sequences is provided in Appendix A.

### 2.10. Western Blot Analysis

Western blot analyses were performed as described previously [23]. Primary antibodies against inducible nitrogen oxide synthase (iNOS) (Abcam, ab178945, Cambridge, UK), cyclooxygenase 2 (COX-2) (Abcam, ab179800, Cambridge, UK), zonula occludens-1 (ZO-1) (Abcam, ab96587, Cambridge, UK), occludin (Abcam, ab167161, Cambridge, UK), signal transducer and activator of transcript 3 (STAT-3) (Abcam, ab68153, Cambridge, UK), p-STAT-3 (Abcam, ab32143, Cambridge, UK), and GADPH (Abcam, ab8245, Cambridge, UK) were used. Protein signals were detected with efficient chemiluminescence (ECL) reagent based on the manufacturer’s instructions.

### 2.11. 16S rDNA Sequencing and Analysis

The genomic DNA was extracted from mice feces samples using the DNA Stool Kit (Tiangen, Beijing, China), according to the protocol. The primers for PCR amplification of the V3–V4 regions of the 16S rRNA genes were: 338F (5′-GTGCCAGCMGCCGCGG-3′) and 806R (5′-CCGTCAATTCMTTTRAGTTT-3′). Amplification was performed on a PCR System (Applied Biosystems, Foster City, CA, USA) with the following reactions: 95 °C for 3 min, 25 cycles at 95 °C for 30 s, 72 °C for 30 s, and a final expansion at 72 °C for 5 min. The amplification products were extracted, purified, and quantified as previously described [23].

Sequencing libraries were created according to the protocols (Illumina, San Diego, CA, USA). Sequencing and data analyses were performed via the Illumina MiSeq platform by a commercial company (Majorbio, Shanghai, China). The raw fastq format sequencing files with a similarity of more than 97% were processed and further classified into operational taxonomic units (OTUs). The bacterial alpha diversity was analyzed using (Mothur, Ann Arbor, MI, USA) (v1.30.1) and is expressed by the Shannon and Chao index [27]. The bacterial beta diversity analysis was evaluated with unweighted UniFrac for Principal Coordinate Analysis (PCoA).

### 2.12. Statistical Analysis

The data are expressed as mean ± standard error of mean (SEM) and were analyzed using GraphPad Prism software (GraphPad, San Diego, CA, USA). Statistical significance among the groups was evaluated by one-way analysis of variance (ANOVA), or two-way ANOVA with Tukey’s multiple comparisons test. The difference of results was considered statistically significant when *p* < 0.05.

## 3. Results

### 3.1. BL Prevents Colitis Symptoms and Tumorigenesis

The cancer-protective effects of BL were examined in mice treated with AOM/DSS (Figure 1A). We observed that the daily food intake and body weight change did not differ among the groups before the AOM/DSS treatment (Appendix A). During the DSS treatment cycles, severe weight loss and increased DAI were observed in mice treated with AOM/DSS (Figure 1B,C). However, BL effectively prevented the body weight loss and improved the DAI score (Figure 1B,C). Colonic shortening is an important hallmark of intestinal inflammation [10]. As shown in Figure 1D, the shortened colon length in mice with colorectal cancer induction was inhibited by BL. Next, tumor formation was analyzed and all of the mice treated with AOM/DSS had developed tumors in the colon. However, fewer tumors and markedly decreased tumor load were found in BL-supplemented mice (Figure 1E,F). Thus, BL can attenuate colitis-associated symptoms and colorectal tumorigenesis.

### 3.2. BL Reduces Inflammation

To study the impacts of BL on inflammatory damage and epithelial disruption, colonic morphology was evaluated by HE staining. Compared to the control mice, colorectal cancer induction resulted in serious inflammatory cell infiltration, distorted crypts, and extensive mucosal damage (Figure 2A), whereas the severity of these pathological features was reduced by BL supplementation (Figure 2A). In addition, mice with colorectal cancer induction had a significant increase in the colonic level of MPO (Figure 2B). BL significantly reduced the MPO level (Figure 2B). To further evaluate the effects of BL on inflammatory responses, the expression levels of inflammatory enzymes were measured. As shown in Figure 2C,D, the elevation of iNOS and COX-2 in colorectal cancer mice was markedly reduced by BL. Quantification of inflammatory cytokines by ELISA revealed that AOM/DSS-treated mice had significant elevation of interferon (IFN)-γ, IL-1β, TNF-α, and IL-6, which could be significantly inhibited by BL supplementation (Figure 2E–H).

### 3.3. BL Improves Colonic Mucosal Barrier Function

The intestinal epithelial barrier is fundamental for modulating intestinal homeostasis [28]. As shown in Figure 3A, gut barrier function was obviously disrupted, as evidenced by the increased intestinal permeability in AOM/DSS-treated mice. Notably, BL-supplemented mice were effectively protected from disruption of the intestinal barrier caused by the AOM/DSS treatment (Figure 3A). Consistently, the reduced levels of ZO-1 and occludin in colorectal cancer mice were significantly attenuated by BL supplementation (Figure 3B,C). Furthermore, AB staining revealed that a significant decrease in goblet cell numbers was triggered by AOM/DSS, whereas BL effectively prevented the depletion of goblet cells (Figure 3D). Consistently, BL also effectively alleviated a reduction in colonic Muc-2 positive cells (Figure 3E). These data indicate that the mucosal barrier dysfunction triggered by AOM/DSS can be improved by BL supplementation.

### 3.4. BL Reduces Cell Proliferation

Colon carcinogenesis is influenced by the aberrant cell proliferation in intestinal epithelium. We subsequently performed Ki67 and proliferating cell nuclear antigen (PCNA) staining to examine the impacts of BL on cell proliferation. As shown in Figure 4A,B, higher levels of Ki67-positive cells and PCNA-positive cells were detected in mice with colorectal cancer induction. However, BL supplementation resulted in an obvious reduction in the number of Ki67- and PCNA-positive cells (Figure 4A,B). These data indicate that BL supplementation can prevent AOM/DSS-induced tumorigenesis by suppressing tumor cell proliferation.

### 3.5. BL Inhibits β-Catenin and STAT3 Signaling

To delineate the underlying mechanisms, we examined the impacts of BL on the β-catenin signaling pathway. The β-catenin level in colonic tissues increased in mice treated with AOM/DSS (Figure 5A). Notably, BL effectively inhibited the significant elevation of β-catenin induced by AOM/DSS (Figure 5A). Furthermore, BL significantly suppressed the mRNA level of β-catenin target genes (Figure 5B). Next, we examined the impacts of BL on the level of STAT3, as STAT3 signaling also participates in mediating proliferation and the survival of tumor cells [12]. As shown in Figure 5C, BL obviously reduced protein levels of phosphorylated STAT3. Overall, these data indicate that BL may attenuate AOM/DSS-induced tumorigenesis, at least in part, by inhibiting β-catenin and STAT3 signaling.

### 3.6. BL Attenuates Dysbiosis of Gut Microbiota

We further examined the changed structure in the gut microbiome. As shown in Figure 6A,B, AOM/DSS induced a reduction in alpha diversity, as revealed by the Chao and Shannon indices. However, BL effectively inhibited the reduced bacterial richness and diversity (Figure 6A,B). Furthermore, a different clustering of bacterial community among the treatment groups was revealed by the PCoA (Figure 6C). Notably, the fecal bacterial composition in mice supplemented with BL showed a higher trend similarity to control mice than AOM/DSS-treated mice without BL supplementation (Permutational multivariate analysis of variance, PERMANOVA, *p* < 0.05) (Figure 6C). Next, the altered proportions of bacterial phylum and genus were further studied. As shown in Figure 6D,E, a reduced abundance of Actinobacteria is detected in mice treated with AOM/DSS when compared to control mice. However, BL obviously increased the proportion of Actinobacteria in AOM/DSS-treated mice (Figure 6D,E). At the genus level, a significant enrichment of *Bifidobacteri**um* was detected in mice supplemented with BL (Figure 6F,G).

## 4. Discussion

It was reported that colitis-associated colorectal cancer has become a major health problem [1]. Although the exact pathogenesis of colitis-associated cancer is still not clear, diet may be an important risk factor that links to the progression of colorectal cancer [29]. Previous findings showed that the gut microbiome, which can be remarkably shaped by diet, is also a key contributing factor to the carcinogenesis of colorectal cancer [15]. Thus, developing dietary agents manipulating the intestinal microbiome is a promising strategy to reduce the global burden of colorectal cancer. This study investigated the role of BL on colitis-associated tumorigenesis and found that dietary BL prevented tumor formation, ameliorated severity of inflammation, improved intestinal barrier function, and attenuated gut microbiota dysbiosis.

The AOM/DSS-induced colon tumorigenesis is a well-established animal model mimicking the etiology of colitis-associated colon cancer in humans [30]. It has been used to study potential pathogenesis and novel therapeutic targets for colorectal cancer. This model develops multistep processes of carcinogenesis characterized by severe colitis-associated symptoms and colorectal tumorigenesis [24]. Our results show that BL attenuates the severity of colitis symptoms as evidenced by the reduced body weight loss, DAI scores, and epithelial damage. BL also inhibits AOM/DSS-induced tumorigenesis by reducing the volume and burden of colon tumors.

Inflammation is a self-limiting physiological process modulated by the balance between anti-inflammatory and pro-inflammatory factors [5]. However, chronic inflammation can cause DNA damage and genetic mutation, and thereby promoting carcinogenesis by stimulating cell proliferation [6,11]. Inflammatory mediators IL-1β, TNF-α, and IL-6 are increased at the early onset of colorectal cancer, and inhibiting the production of these cytokines can be an effective strategy to prevent colon cancer development [31]. Our results also reveal that AOM/DSS-treated mice had elevated levels of TNF-α, IL-6, IFN-γ, and IL-1β, which were effectively reduced by dietary BL supplementation.

The persistent chronic inflammation induced by the administration of DSS can also damage the colonic mucosa and further cause injury to intestinal epithelium. The abundant mucus protein produced by goblet cells forms an intestinal mucus layer that can protect against the invasion of pathogens [32]. However, a defective intestinal mucus barrier was suggested to promote the progression of colonic tumorigenesis [33]. Intestinal tight junction proteins are thought to be crucial for maintaining intestinal homeostasis [34]. The progression of colon cancer is closely related to decreased occludin and ZO-1 [27,35]. Our data show that BL mitigated AOM/DSS-induced depletion of colonic goblet cells. Moreover, elevation of occludin and ZO-1 was observed after BL supplementation. Thus, it is likely that the cancer-protective effects of BL are related to the enhancement in intestinal barrier functions.

STAT3, a key signaling pathway that regulates cell growth, proliferation, and survival, functions as an important mediator linked to chronic inflammation and cancer [36]. Studies reported that its activation can be induced at the inflammatory sites and can promote the transcription of the specific genes involved in carcinogenesis [36]. Moreover, β-catenin is also a well-studied signaling pathway that plays critical roles in tumorigenesis [37,38]. Our data show that the AOM/DSS-induced increases in β-catenin signaling and the phosphorylated form of STAT3 were effectively inhibited by BL, suggesting that the beneficial effects of BL against colon cancer development may partially depend on the inhibition of STAT3/β-catenin signaling.

The gut is inhabited by a complex microbial community, which influences various physiological functions of the host. The disruption of the intestinal microbial community has been reported to be correlated with the pathogenesis of colorectal cancer [14,15,39,40]. The expansion of several specific bacterial strains including *Escherichia coli**, Fusobacterium nucleatum*, and *Bacteroides fragilis* were reported to be associated with colorectal carcinogenesis [17]. In addition, reduced bacterial diversity is also considered a risk factor contributing to the increased incidence of colorectal cancer [41]. Our microbiota sequencing analysis showed that AOM/DSS induced a dramatic reduction in bacterial diversity. It is noted that the relative abundance of *Bifidobacteria*, which was reported to have antitumor effects [42], was significantly enriched in mice supplemented with BL. It was reported that increased consumption of plant fiber can help create a healthy intestinal microbial community by promoting beneficial bacterial growth and proliferation [43]. Thus, the enrichment of *Bifidobacteria* may result from the degradation of the fiber component in BL, which counterbalances the AOM/DSS-induced dysbiosis of gut microbiota. However, we cannot rule out the possibility that other bioactive components of BL may also have an active impact on the regulation of gut microbiota [21]. Consistent with our results, fermented barley also had protective roles against DSS-induced acute colitis through increasing the abundance of *Lactobacillus* [44]. Our data indicate that barley and BL may be developed as prebiotic agents to attenuate other inflammatory diseases by targeting the gut microbiota [45]. Further research is needed to clarify whether the enrichment of *Bifidobacterium* is due to the direct action of BL on intestinal bacteria or the effect of BL on the host physiology, which subsequently affects the gut microbiome.

## 5. Conclusions

In conclusion, BL effectively attenuated colon tumorigenesis, and the underlying mechanisms were linked to the amelioration of inflammation and prevention of microbiota dysbiosis. BL can be used as a potent dietary supplement for the prevention of colorectal carcinogenesis.

## Figures and Tables

**Figure 1 nutrients-13-03487-f001:**
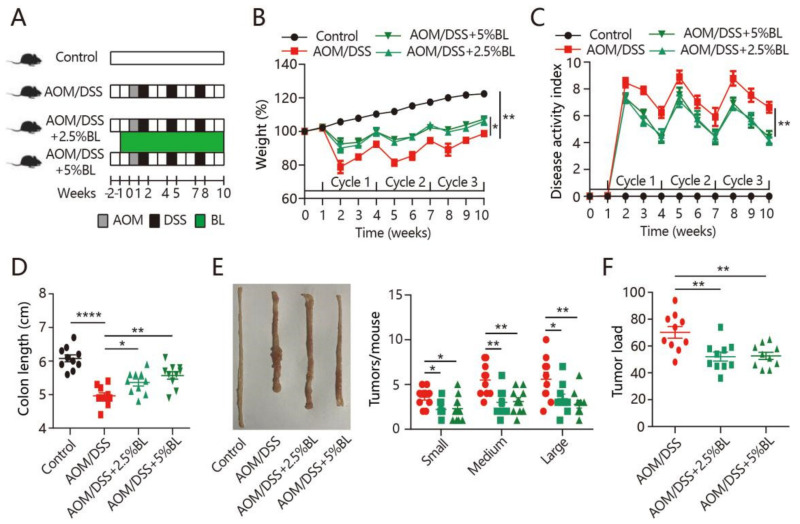
BL attenuates AOM/DSS-induced colorectal cancer. (**A**) Experimental protocol for intervention timeline and induction of colitis-associated colon carcinogenesis. (**B**) Percentage weight change, (**C**) disease activity index, (**D**) colon length, (**E**) representative morphologies of colon and tumor number, and (**F**) tumor load from different mouse groups were measured (*n* = 10). Data are expressed as mean ± SEM. (**B**,**C**) Two-way ANOVA. (**D**–**F**) One-way ANOVA. * *p* < 0.05, ** *p* < 0.01, and **** *p* < 0.0001. BL, barley leaf; AOM, azoxymethane; DSS, dextran sodium sulfate.

**Figure 2 nutrients-13-03487-f002:**
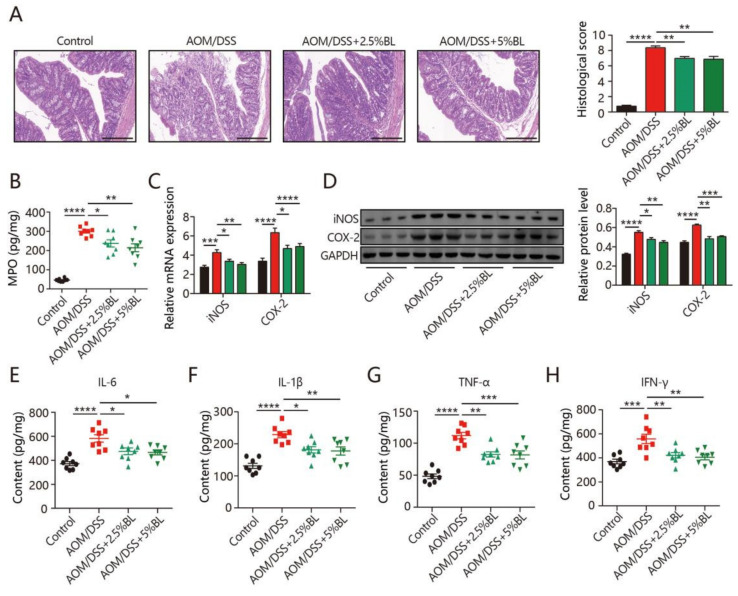
BL inhibits AOM/DSS-induced inflammatory mediators. (**A**) The histological score and images of HE staining in colon (400×). (**B**) The level of myeloperoxidase (MPO). (**C**,**D**) The levels of inducible nitrogen oxide synthase (iNOS) and cyclooxygenase 2 (COX-2), and (**E**–**H**) the levels of interleukin (IL)-6, IL-1β, tumor necrosis factor (TNF-α), and interferon (IFN)-γ in the colonic tissues (*n* = 8–10). Data are mean ± SEM. * *p* < 0.05, ** *p* < 0.01, *** *p* < 0.001, and **** *p* < 0.0001. BL, barley leaf; AOM, azoxymethane; DSS, dextran sodium sulfate.

**Figure 3 nutrients-13-03487-f003:**
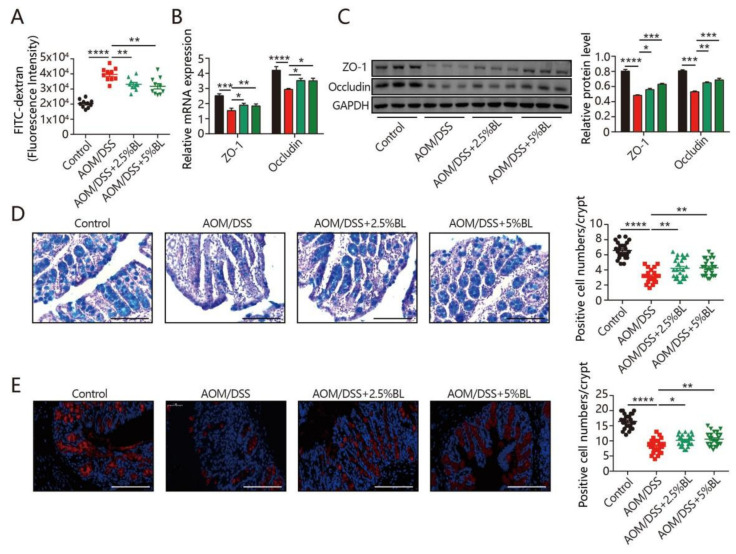
BL protects against AOM/DSS-induced intestinal barrier dysfunction. (**A**) Intestinal permeability, and (**B**,**C**) the expression of zonula occludens-1 (ZO-1) and occludin in the colonic tissues (*n* = 8–10). (**D**) AB staining and colonic goblet cell numbers were quantified (400×). (**E**) Immunofluorescent analysis of Muc-2 (red) in mouse colonic sections (400×). Data are mean ± SEM. * *p* < 0.05, ** *p* < 0.01, *** *p* < 0.001 and **** *p* < 0.0001. The statistics were performed with one-way ANOVA. BL, barley leaf; AOM, azoxymethane; DSS, dextran sodium sulfate.

**Figure 4 nutrients-13-03487-f004:**
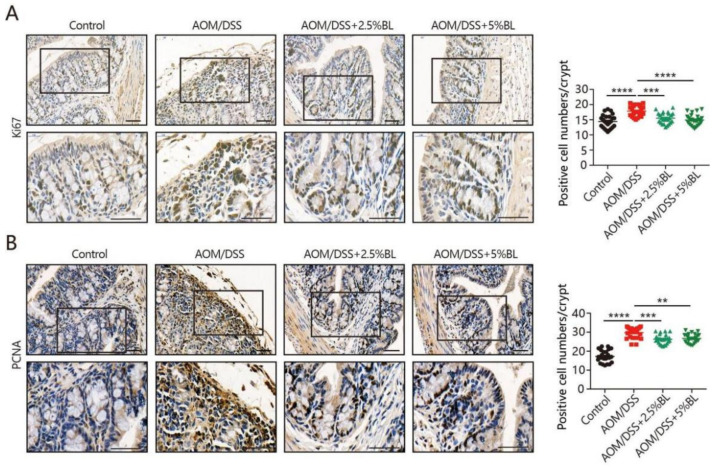
BL suppresses AOM/DSS-induced cell proliferation. (**A**) Immunohistochemistry analysis of Ki-67 in mouse colonic sections and positive cell numbers were quantified. Upper: (200×). Lower: (400×). (**B**) Immunohistochemistry analysis of proliferating cell nuclear antigen (PCNA) in mouse colonic sections and positive cell numbers were quantified. Upper: (200×). Lower: (400×). Data are mean ± SEM. ** *p* < 0.01, *** *p* < 0.001, and **** *p* < 0.0001. The statistics were performed with one-way ANOVA. BL, barley leaf; AOM, azoxymethane; DSS, dextran sodium sulfate.

**Figure 5 nutrients-13-03487-f005:**
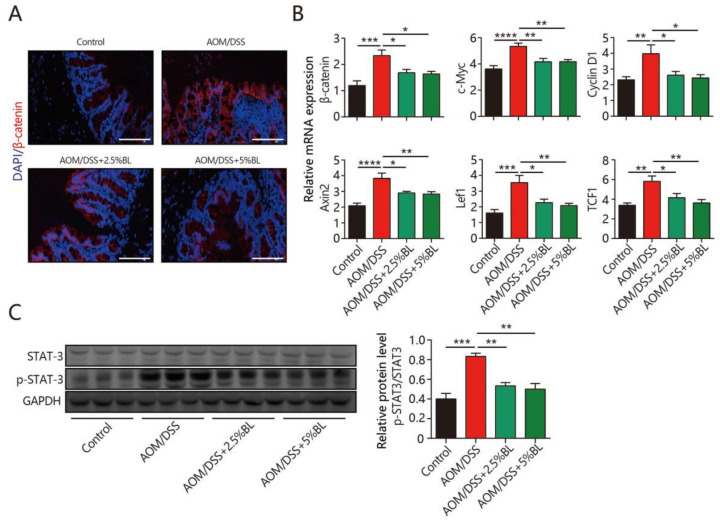
BL inhibits AOM/DSS-induced β-catenin signaling activation. (**A**) Immunofluorescent analysis of β-catenin (red) in mouse colonic sections (400×). (**B**) The expression of target genes in β-catenin signaling pathway and (**C**) protein levels of STAT3 and p-STAT3 in the colonic tissues were measured (*n* = 8–10). Data are mean ± SEM. * *p* < 0.05, ** *p* < 0.01, *** *p* < 0.001, and **** *p* < 0.0001. The statistics were performed with one-way ANOVA. BL, barley leaf; AOM, azoxymethane; DSS, dextran sodium sulfate.

**Figure 6 nutrients-13-03487-f006:**
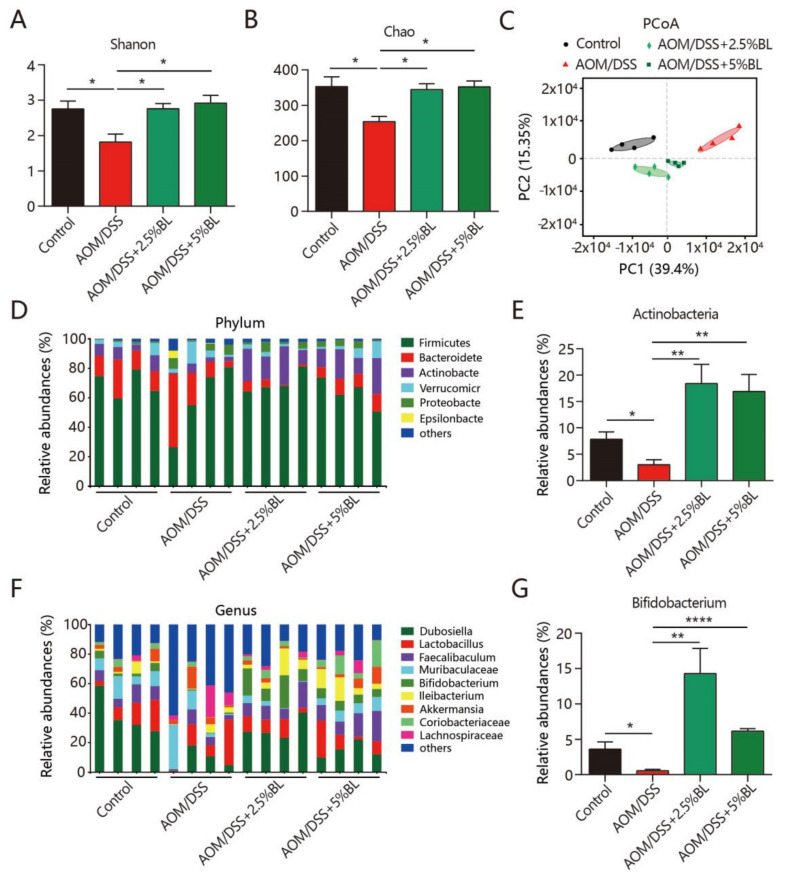
BL improves AOM/DSS-induced gut microbiota dysbiosis. (**A**) Shannon index and (**B**) Chao index. (**C**) PCoA of the fecal bacterial composition. (**D**) Distributions of bacterial community at phylum level and (**E**) the relative abundance of Actinobacteria. (**F**) Distributions of bacterial community at genus level and (**G**) the relative abundance of Bifidobacterium. * *p* < 0.05, ** *p* < 0.01, and **** *p* < 0.0001. The results are expressed as mean ± SEM. BL, barley leaf; AOM, azoxymethane; DSS, dextran sodium sulfate.

## Data Availability

The number of the sequencing project is PRJNA702637.

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
