# Peer review of "Dietary Barley Leaf Mitigates Tumorigenesis in Experimental Colitis-Associated Colorectal Cancer"

_nutrients, 2021, doi:10.3390/nu13103487_

Round 1

Reviewer 1 Report

This manuscript examines the effects of barley leaf extract against colitis-associated colorectal carcinogenesis in mice. Overall, this manuscript looks good although it can be improved on certain aspects. I have few questions for the authors summarized below:

  • It is not clear how many males and females were present in each of the four mice groups. Were all the groups consisting of males and females of equal numbers? The authors should clarify this.
  • The authors should explain in more details about the 16S PCR primers and cite them properly if taken from a previous study.
  • The authors can add details about the filters and parameters which were used in data cleaning steps and if they used mothur for this purpose.
  • The authors should also properly cite the mothur software and mention its version which was used here.
  • It appears that the results are quite similar for 2.5% BL and 5% BL as evident from figures 2 and 3. Does it suggest that the effects of BL are not that effective beyond a certain level even if BL dose is increased? What is the opinion of authors on this and how do they think it can be improved?
  • From the manuscript it is not clear if the BL is directly affecting the tumorigenesis. It is possible that all the effects are due to its influence on colitis which is associated with the colorectal cancer. What is the opinion of authors on this?
  • Figures 6D and 6F does not add much information. Large variability can be seen even within the groups and the color codes are not explained in the legend for the readers to identify the phylum or genus from the figures. I think a table with relative abundances of major phyla and genera might be more useful.
  • It would be more helpful if the authors find the bacteria most differentiating across the groups rather than just targeting Bifidobacterium. For example, did they observed Akkermansia in their data and how did it vary across the groups?
  • The authors can explain more how the findings of this work can be implemented in future studies and what precautions other research groups should take if they want to continue or take this work to the next level.

Reviewer 2 Report

Re: Manuscript ID: nutrients-1394284.

I congratulate the authors on this well written research article dealing with the protective effect of dietary barley leaf in an experimental model of colitis-associated colorectal cancer in mice. The importance of this topic is growing and new aspects have been described in the manuscript. Some changes are suggested to improve the paper.

Points of criticism

How were the animal sacrificed? Please, describe.

Line 25. Replace “microibota” with “microbiota”.

Line 80. Replace “prevent” with “prevented”.

Line 193. Replace “were” with “was”.

The caption of Figure S1 must report that data refer prior to the treatment of AOM/DSS, as stated in line 200 of the manuscript.

Line 209. Replace “suppress” with “reduce” or “attenuate” (as in line 212 of caption of fig. 1). Indeed, the treatment does not completely reverse the colitis-associated symptoms and colorectal tumorigenesis induced by AOM/DSS. See also lines 257, 265-266, 277, 307, 362, 383.

Line 212. Replace “azoxymethane (AOM)/dextran sulfate sodium (DSS)” with “AOM/DSS”. The same change in the subsequent figures.

Unlike fig. 1, the font size of captions of figs. 2-6 is the same of the manuscript text.

Line 238. Replace “(Figure 2E-F)” with “(Figure 2E-H)”.

Figure 3D-E. Y-axis: crept? (maybe crypt).

Line 278. Replace “of in” with “in”.

Line 285. Replace “(Figure 1D and E)” with “(Figure 1E and F)”.

Figure 4. Y-axis: crept? (maybe crypt). Check the scale bar.

Line 323. Replace “index were” with “index which were”.

Lines 427-428. Replace “possibility other” with “possibility that other”.

Line 494: 2015 in bold.

I suggest the authors to consider and comment also these articles:

Li D, Wang P, Wang P, Hu X, Chen F. Gut microbiota promotes production of aromatic metabolites through degradation of barley leaf fiber. J Nutr Biochem. 2018 Aug;58:49-58. doi: 10.1016/j.jnutbio.2018.05.001.

Woo JK, Choi S, Kang JH, Kim DE, Hurh BS, Jeon JE, Kim SY, Oh SH. Fermented barley and soybean (BS) mixture enhances intestinal barrier function in dextran sulfate sodium (DSS)-induced colitis mouse model. BMC Complement Altern Med. 2016 Dec 3;16(1):498. doi: 10.1186/s12906-016-1479-0.

Please, consider also protective effects exerted by barley leaf in other organs, for example:

Hwang YH, Ha H, Kim R, Cho CW, Song YR, Hong HD, Kim T. Protective effects of a polysaccharide BLE0 isolated from barley leaf on bone loss in ovariectomized mice. Int J Biol Macromol. 2019 Feb 15;123:314-321. doi: 10.1016/j.ijbiomac.2018.11.075.

Reviewer 3 Report

The authors of the manuscript " Dietary barley leaf ...cancer" is a good attempt to mitigate experimental colitis associated colorectal cancer.  The authors tried but there are few fundamental questions:

  1. Please provide a clear timeline of the experimental protocol in the methods section. Though the authors have given a schematic diagram, the authors should write about the same to clarify.
  2. On the same line its not clear when was the BL (5% or 2.5%) given to the animals.  Was it given 1 week before the experiments started or what timeline..  Please make a method section of it.
  3. No where in the experiments have the authors given the exact end point.  What day post experiment was the animals sacrificed or if there was any other interventions done.  Please clarify.
  4. the authors should provide in what was the protein extract made (RIPA) and how much protein was used for electrophoresis.
  5. the authors should provide with the photographs of the tumor in the colon which would make the paper more interesting.  though the authors have given graph of tumor load but pics of the colon would be great.
  6. In section 3.2 the authors state that AOM/DSS mice exhibit distorted crypts and extensive mucosal damage is not at all clear from the histological pics provided.  The authors should include the pics of the crypts long with the vili length.  How was the MPO measured is not given in the methods section and why did the authors want to measure MPO.
  7.  In the section 3.3 the authors state that Consistently, the reduced mRNA and protein levels of ZO-1 and Occludin ..in AOM/DSS treated mice.  But the mRNA levels of both ZO1 n Occludin is upregulated but in the protein levels its reduce.  Can the authors discuss about the same. Also why authors choose ZO-1 and Occludin as there are many publications which show other tight junctions to be involved in intestinal permeability.  the authors did not mention the methods used for intestinal permeability.
  8. Alcian blue and Muc2 both results show the same thing, so why did the authors want to show both IHC results any reason for the same. Also in the graph in Fig 3D and 3E its written positive cell numbers/crept.  Why is that so? the spelling is crypt. 
  9. Similarly the same in proliferation experiments why the authors used Ki67 and PCNA to show the same results.  Why the authors think decreasing proliferation is good for the animals?  The increase in proliferation is good as it will help block the intestinal leak and not the vice versa. Any good suggestions.

Round 2

Reviewer 3 Report

The authors did all the required changes and the manuscript is ready to be accepted for publication